# Synergistic Microbial Inhibition and Quality Preservation for Grapes through High-Voltage Electric Field Cold Plasma and Nano-ZnO Antimicrobial Film Treatment

**DOI:** 10.3390/foods12234234

**Published:** 2023-11-23

**Authors:** Juan Li, Guantao Zhang, Zitong Zhang, Yuan Zhang, Dongjie Zhang

**Affiliations:** 1College of Food, Heilongjiang Bayi Agricultural University, Daqing 163319, China; lijuan403@163.com (J.L.); gt199801@163.com (G.Z.); zzt1113978282@163.com (Z.Z.); zhangyuan246795@163.com (Y.Z.); 2National Coarse Cereals Engineering Research Center, Daqing 163319, China

**Keywords:** grapes, high-voltage electric field cold plasma, ZnO nanoparticles, antimicrobial coating, synergistic effect, sterilization and preservation, food safety

## Abstract

To ensure their quality and safety, harvested grapes should be protected from microbial contamination before reaching consumers. For the first time, this study combined high-voltage electric field cold plasma (HVEF-CP) and nano-ZnO antimicrobial film to inhibit microbial growth on grapes. Using the response surface method, the optimal processing parameters of HVEF-CP (a voltage of 78 kV, a frequency of 110 Hz, and a time of 116 s) were identified to achieve 96.29% sterilization. The effects of co-processing with HVEF-CP and nano-ZnO antimicrobial film on the quality and safety of grapes during storage were explored. When stored at 4 °C and 20 °C, the co-processing extended the shelf life of grapes to 14 and 10 days, respectively. The co-processing increased the sterilization rate to 99.34%, demonstrating a synergistic effect between the two methods to ensure not only the safety of grapes but also their nutrient retention during storage. This novel approach is promising for the efficient, safe, and scalable preservation of grapes as well as other foods.

## 1. Introduction

During transportation and storage, fresh grapes are prone to different types of damage, such as threshing, stem drying, and browning. In addition, bacteria, yeast, and mold on the grapes’ surface may reduce their safety, quality, and nutritional value [1,2]. Current methods to preserve and sterilize grapes have employed shortwave ultraviolet (UV) irradiation, ultrasound, and ozone. However, the sterilization efficiency is low and limited by the shape and stacking placement of the fruits. Meanwhile, traditional chemical sterilization treatments can leave behind residues that are detrimental to human health [3,4]. Therefore, it is crucial to develop efficient, green, safe, and scalable technologies for the sterilization and preservation of grapes.

A recent trend in food sterilization is a transition from traditional “hot processing” to “cold processing” (non-thermal) technologies [5]. The combination of high-voltage electric field cold plasma (HVEF-CP) technology and modified atmosphere packaging with gas filling has been applied as a non-thermal sterilization and environmentally friendly technique [6]. HVEF-CP produces various active substances with excellent sterilization effects, such as reactive oxygen, reactive nitrogen, charged particles, and UV photons [7,8]. These active substances interact with surface microbes to affect the cell structure and inactivate the microbes [9,10,11]. Due to its low energy consumption, high antibacterial effect, absence of pollution or residue, high efficiency, and rapid deployment [12,13], HVEF-CP is widely used for fresh fruits and vegetables [14,15] as well as meat products [16,17,18,19]. This technology has been reported to effectively eliminate pathogenic microbes [20], fungi, spores [21], and viruses [22], as well as degrading pesticide residues. However, the antimicrobial performance of HVEF-CP is affected by the processing parameters, ambient factors, and microbial attributes, among others [23]. The device used to generate HVEF-CP in this study, plasma formation mechanism, and sterilization mechanism are shown in Figure 1.

ZnO nanoparticles have demonstrated antibacterial effects on foods [24]. The mechanism of nano-ZnO involves contact, permeation, and oxidation, which affect the microbial cell membranes and internal components, and ultimately render the microbes inactive [25,26]. However, the sterilization effect can be reduced by insufficient ambient light [27], excessive ZnO particle size [28], and the presence of oxygen vacancies [29].

To the best of our knowledge, there has been no report of simultaneously using HVEF-CP and nano-ZnO on grapes or the impact of such co-processing on grape quality. The combination of these two methods can overcome their individual limits, assure grape quality and safety during transportation and storage, maximize nutrient retention, and extend the shelf life. Therefore, this study examined the antimicrobial performance of the combined approach (HVEF-CP + nano-ZnO) as well as its effects on grape quality and shelf life. The results provide useful guidance for developing efficient, safe, and scalable grape preservation methods. There is also broader application potential for other perishable foods.

## 2. Materials and Methods

### 2.1. Raw Materials and Chemicals

Grapes of the Kyoho variety were grown at Grape Ecological Park (Beizhen, Jinzhou City, Liaoning Province, China). Nutrient broth (NB) was obtained from Beijing Aoboxing Biotechnology. Nutrient agar (NA) medium, potato dextrose agar (PDA) medium, and potato dextrose broth (PDB) were purchased from Qingdao Haibo Biotechnology (Qingdao, China). Further, 2,6-Dichloroindophenol was obtained from Shanghai Yansheng Industrial, and ascorbic acid was obtained from Shanghai Sinopharm Chemical Reagent (Shanghai, China).

Microbe samples (*Escherichia coli*, *Staphylococcus aureus*, *Pichia pastoris*, and *Aspergillus niger*) were obtained from Heilongjiang Bayi Agricultural University (Daqing, China).

### 2.2. Apparatus and Analytical Tools

HVEF-CP was generated using a CPS-I high-voltage electric field low-temperature plasma cold sterilization test system (Nanjing Yirun Plasma Technology, Nanjing, China). The grapes were packaged using a DT-6D-controlled atmosphere fresh-keeping packaging machine (Shanghai Zhonglin Mechanical and Electrical Equipment, Shanghai, China). Incubation was carried out in an SPX-150B biochemical incubator (Tianjin Hongnuo Instruments, Tianjin, China).

The other tools and instrument used in this study included: LHS-2413 portable pressure steam sterilizer (Ningbo Linghong Medical Instruments, Ningbo, China), Scientz-04 slap homogenizer (Ningbo Xinzhi Biotechnology, Ningbo, China), 101-2A electric blast drying oven (Zhengzhou Preston Technology, Zhengzhou, China), BMH-13 spiral micrometer (Jinan Languang Electromechanical Technology, Jinan, China), YP200001 electronic balance (Shanghai Yueping Scientific Instruments, Shanghai, China), PAL-ACID1 mini digital acidity meter (Japan Atago Electrochemical Instruments, Tokyo, Japan), and SPECORD^®^ 210 PLUS UV-visible spectrophotometer (Jena Analytical Instruments, Jena, Germany).

### 2.3. Methods

#### 2.3.1. Sample Groups

The grapes were divided into five groups: control group (untreated and unpacked), polypropylene (PE) film group (packed with PE film), HVEF-CP group (unpacked and treated with HVEF-CP), nano-ZnO group (packed with nano-ZnO antimicrobial film), and HVEF-CP + nano-ZnO group.

#### 2.3.2. Co-Processing of Grapes

The selected grapes were at similar stages of maturity and free of mechanical damage, pests, or diseases. They were inoculated with a fixed amount of *S. aureus* (logarithm of the total number of colonies: about 3.86 lg (CFU·g^−1^)). To apply the nano-ZnO antimicrobial film, 10 grape berries were placed in a bag made of nano-ZnO antimicrobial film and then sealed on all sides [30]. The sealed bag was placed in a polypropylene (PP) cartridge (size: 195 mm × 135 mm × 40 mm),which was filled with air and sealed. The sealed cartridge was placed between the two polar plates of the HVEF-CP system for sterilization under specific processing parameters (voltage, frequency, and time). After treatment, each sample was incubated at 4 °C (for 14 days) or 20 °C (for 10 days) under 60% humidity, and then the microbial and physiochemical indices were determined, including the lg (CFU·g^−1^) value, total soluble sugar (TSS) content, vitamin C (VC) content, total acid (TA) content, and mass loss rate. Each experimental measurement was repeated three times. The process flowchart for the synergistic treatment of grapes with HVEF-CP and nano-ZnO film is shown in Figure 2.

#### 2.3.3. Optimizing the HVEF-CP Parameters

##### Single-Factor Experiments

The processing voltage, frequency, and time were selected as three single factors to identify the optimal parameters for cold sterilization of grapes by HVEF-CP. The antibacterial performance was evaluated in terms of lg (CFU·g^−1^), as detailed in Table 1.

##### Response Surface Experiment

Based on the results of the single-factor tests, the Box–Behnken experimental design implemented in Design-Expert 13.0 (Stat-Ease Inc., Minneapolis, MN, USA) was employed to further optimize the HVEF-CP parameters. The processing voltage (X_1_), processing frequency (X_2_), and processing time (X_3_) were used as independent variables, and the lg (CFU·g^−1^) value was used as the response value (Y), as detailed in Table 2.

#### 2.3.4. Measurement Methods

The antibacterial performance of nano-ZnO antimicrobial films was measured according to Wang et al. [31]. We considered four microbial species (*E. coli*, *S. aureus*, *P. pastoris*, and *A. niger*) because they are all closely associated with grape spoilage. After isolating and purifying the test strains, they were evenly spread on the target culture medium. Subsequently, a disc of antimicrobial film (10 mm in diameter) was placed on the surface of the inoculated culture medium. The petri dish was inverted and incubated at 37 °C for 24 h (for *E. coli* and *S. aureus*) or at 28 °C for 2–5 d (for *P. pastoris* and *A. niger*) in a CO_2_ incubator. Afterwards, the diameter of the inhibition zone was measured using a caliper. The lg (CFU·g^−1^) value and sterilization rate were calculated according to GB 4789.2-2016 [32].

The TSS content was determined using an Abbe refractometer. The VC content was determined using the 2,6-dichloroindophenol method [33], and the results are given in units of (mg/100 g). The TA content was measured using a mini digital acidity meter. All tests were conducted with three parallel samples. The mass loss rate was calculated using Equation (1) [34]:(1)$$Mass\;loss\left(\%\right)=\frac{Mass\;before\;storage\;-\;Mass\;after\;storage}{Mass\;before\;storage}\times 100\%$$

#### 2.3.5. Statistical Analysis

Each group was tested three times in parallel, and SPSS 19.0 (IBM, Armonk, NY, USA) was used for data analysis and processing. Duncan’s multiple range test was used for significance detection, in which different letters (a–f and A–F) denote significant differences in the same indicator under the same conditions (*p* < 0.05). Origin 9.0 (OriginLab, Northampton, MA, USA) was used for plotting. Results from the response surface test (*p* < 0.05) were analyzed by Design-Expert 13.0 (Stat-Ease, Minneapolis, MN, USA).

## 3. Results and Discussion

### 3.1. Performance of Nano-ZnO Antimicrobial Film

The antimicrobial mechanism of nano-ZnO is shown in Figure 3. ZnO exhibits strong antimicrobial activity against a broad spectrum of pathogens [35]. Upon contact with microbes in the culture medium, the ZnO particles kill microbes by interacting with the microbial surface. The nano-ZnO antimicrobial film generates reactive oxygen species (ROS) such as hydroxyl radicals (HO·) and superoxide radicals (·O_2_^−^) under visible light irradiation. These species can destroy cell membranes and cell walls, leading to the death of microbes. Nano-ZnO also gradually releases antibacterial Zn^2+^ into bacteria, destroying the cell membrane and denaturing proteins and nucleic acids to diminish the cell’s ability to proliferate, ultimately causing cell death [36].

The diameter of the inhibitory zone is conventionally used to describe the antimicrobial effect. According to the results in Table 3 and Figure 4, the nano-ZnO antimicrobial film inhibited all four studied microbial species, with the sensitivity ranked as *S. aureus* > *E. coli* > 15 mm (highly sensitive) > *A. niger* > *P. pastoris* > 10 mm (moderately sensitive). Therefore, we selected *S. aureus* as the strain for the single-factor experiments.

### 3.2. Optimization of HVEF-CP Processing Parameters

#### 3.2.1. Single-Factor Experimental Results

##### Effect of Processing Voltage

As shown in Figure 5, the lg (CFU·g^−1^) value of *S. aureus* significantly decreased when the processing voltage increased while the sterilization rate significantly increased (*p* < 0.05). This result is consistent with that reported by Liu et al. [37] because a higher voltage excites air in the PP cartridge to produce more active substances. After processing at the lowest voltage of 55 kV, the lg (CFU·g^−1^) value dropped from the initial 3.86 to 3.32, indicating a sterilization rate of 70.91%. This performance was poor because the low voltage failed to produce a sufficient quantity of active substances. When the processing voltage rose to 75 kV, the lg (CFU·g^−1^) value was reduced to 2.35 and the sterilization rate reached 96.88%. However, at even higher processing voltages (85–95 kV), the lg (CFU·g^−1^) value increased again to 2.67, while the sterilization rate dropped to 93.31% and the change was significant (*p* < 0.05). A voltage exceeding 75 kV caused a strong discharge of the HVEF-CP equipment, which probably led to damage of the PP cartridge and leaking of the active substances, thereby reducing the sterilization rate. Research by Yi and colleagues showed that HVEF-CP treatment at 75 kV significantly inhibits the total microorganism growth and enhances food safety [38]. Based on these results, a processing voltage of 75 kV was selected as the median level for the response surface test, and the processing voltage range was 65–85 kV.

##### Effect of Processing Frequency

As shown in Figure 6, when the processing frequency increased, the lg (CFU·g^−1^) value decreased first and then gradually increased. The lg (CFU·g^−1^) value decreased significantly (*p* < 0.05) with the processing frequency at below 95 Hz. This is consistent with the results of Wei et al. [39]. Compared with the control group, HVEF-CP processing at 55 Hz reduced the lg (CFU·g^−1^) value from 3.86 to 3.08 with an 83.25% sterilization rate. The minimum lg (CFU·g^−1^) value of 2.48 and the maximum sterilization rate of 95.79% were observed at 95 Hz. In the frequency range of 115–135 Hz, the lg (CFU·g^−1^) value rose to 2.65 with a 93.73% sterilization rate, and there was no significant difference between the processed groups (*p* < 0.05). Compared with the lg (CFU·g^−1^) value at 95 Hz, that at higher frequencies increased significantly, and the sterilization rate decreased significantly (*p* < 0.05). A higher discharge power resulted in a higher concentration of active substances, subsequently enhancing the bactericidal efficacy of HVEF-CP [40]. Hence, the processing frequency of 95 Hz was selected as the median level for the response surface test, and the processing frequency range was 65–85 kV.

##### Effect of Processing Time

As shown in Figure 7, when the processing time increased, the lg (CFU·g^−1^) value decreased first and then gradually increased. Meanwhile, the sterilization rate increased first, stabilized, and then decreased slowly. When the processing time was less than 120 s, the sterilization rate was significantly enhanced at longer times (*p* < 0.05) which is consistent with the results of Choi et al. [41]. During processing, the air in the PP cartridge was continuously excited to produce more active substances with antibacterial effects, reducing the lg (CFU·g^−1^) value and enhancing the sterilization effect. Compared with the control group, a processing time of 60 s reduced the lg (CFU·g^−1^) value to 3.19 (78.41% sterilization rate). A minimum plasma exposure time was required to generate enough active substances from excited air in the PP cartridge, and a period of 60 s seemed to be insufficient at the fixed processing voltage and frequency of 75 kV and 95 Hz, respectively. When the processing time increased to 120 s, the lg (CFU·g^−1^) value dropped dramatically to 2.55 and the sterilization rate reached the maximum of 95.04%. On the other hand, when the processing time exceeded 120 s (150–180 s), the lg (CFU·g^−1^) value increased and the sterilization rate decreased. Prolonged discharge processing may have caused invisible damage to the PP cartridge, resulting in the leakage of active substances and reducing the sterilization effect. This aligns with the findings of Zhuang et al. [42]. Therefore, a processing time of 120 s was selected as the median level of the response surface test, with a range of 90–150 s.

#### 3.2.2. Response Surface Test

##### Experimental Design and Results

Based on the single-factor test results, the Box–Behnken Design method implemented in Design-Expert 13.0 was used to optimize the processing parameters for the sterilization of grapes by HVEF-CP, as shown in Table 4.

The lg (CFU·g^−1^) value (Y) was assumed to be a quadratic function of X_1_, X_2_, and X_3_. Polynomial regression of the data in Table 4 gave the following equation:Y = 2.32 − 0.3137 X_1_ − 0.07002 X_2_ + 0.1012 X_3_ − 0.0275 X_1_X_2_ + 0.0400 X_1_X_3_
− 0.0275 X_2_X_3_ + 0.2735 X_1_^2^ + 0.4410 X_2_^2^ + 0.3585 X_3_^2^.

According to Table 5, the model resulted in F = 21.25 and *p* = 0.0003 < 0.01. The difference is therefore extremely significant with an error of only 0.03%. The lack of fit test found that F = 1.04 and *p* = 0.4663 > 0.05, indicating that the difference is not significant. The parameters of CV = 4.27%, R^2^ = 0.9647, and R^2^Adj = 0.9193 show that Y = 91.93% response change. Therefore, the developed model can be used for further analysis.

As shown in Table 5, the differences in X_1_, X_1_^2^, X_2_^2^, and X_3_^2^ are extremely significant (*p* < 0.01), and that in X_3_ is significant (*p* < 0.05). The F-value intuitively indicates the effects of X_1_, X_2_, and X_3_ on the cold sterilization of grapes, and the effects can be ranked as X_1_ (54.12) > X_3_ (5.64) > X_2_ (2.69).

##### Two-Factor Interaction Analysis

Table 5 also shows that the *p* value of the interaction term X_1_X_2_ = X_2_X_3_ = 0.6622 is larger than both X_1_X_3_ = 0.5284 and X_3_ = 0.0493, indicating that they are not significant. The effect of interactions among X_1_, X_2_, and X_3_ on the lg (CFU·g^−1^) value (Y) is as follows.

According to Figure 8, when the processing time of HVEF-CP was fixed at 120 s, increasing the processing voltage caused the lg (CFU·g^−1^) value to first decrease and then slowly increase. When the processing voltage was fixed, increasing the processing frequency produced a similar trend. When the processing voltage and frequency were within 73–80 kV and 105–118 Hz, respectively, the lg (CFU·g^−1^) value reached the minimum and the sterilization rate reached the maximum, suggesting that these processing voltage and frequency together produced the best sterilization effect.

As shown in Figure 9, when the processing frequency was fixed at 95 Hz, increasing either the processing voltage or the processing time caused the lg (CFU·g^−1^) value to first decrease and then slowly increase, forming a quadratic surface. When the processing voltage was 75–82 kV and the processing frequency was 113–118 Hz, the minimum lg (CFU·g^−1^) value and the maximum sterilization rate were realized.

A similar trend is shown in Figure 10 when the processing voltage was fixed at 75 kV while the processing frequency and processing time were varied. The minimum lg (CFU·g^−1^) value and the maximum sterilization rate were reached at 106–110 Hz and 112–120 s, respectively.

##### Determination and Verification of the Optimal Processing Parameters

Analysis using Design-Expert 13.0 yielded the following optimal HVEF-CP parameters: a processing voltage of 78.416 kV, a processing frequency of 109.659 Hz, and a processing time of 115.778 s. Considering the controllability, in the actual experiments these parameters were rounded to 78 kV, 110 Hz, and 116 s, respectively. Based on three replicated experiments, the average lg (CFU·g^−1^) value was 2.43, and the average sterilization rate of 96.29% was similar to that estimated from the response surface test (96.41%). Therefore, it was appropriate to round up the HVEF-CP parameters for the cold sterilization of grapes.

### 3.3. Antibacterial Effect and Quality of Grapes Co-Processed by HVEF-CP + Nano-ZnO Antimicrobial Film

#### 3.3.1. Effects of Different Sterilization Approaches on lg (CFU·g^−1^) Value

In Figure 11, *S. aureus* continuously grew on the grape surface during storage at both 4 °C and 20 °C. The lg (CFU·g^−1^) values in the three sterilized groups after 2 and 4 days were significantly lower than that of the control group (*p* < 0.05). According to Lee et al. [43], consuming foods contaminated with more than 6 lg (CFU·g^−1^) of *E. coli* can cause illness in humans. At 0 days, the lg (CFU·g^−1^) values in the PE film, nano-ZnO, HVEF-CP, and HVEF-CP + nano-ZnO groups were 3.86, 3.19, 2.43, and 1.68, respectively. When used alone, the sterilization rates of nano-ZnO and HVEF-CP groups were 78.62% and 96.28%, respectively. In comparison, the samples treated with both methods achieved a much higher sterilization rate of 99.34%, suggesting more severe retardation of bacterial growth. For the control group stored at 4 °C and 20 °C, the lg (CFU·g^−1^) value exceeded 6 lg (CFU·g^−1^) after 6 and 4 days, respectively. The PE film, nano-ZnO, and HVEF-CP groups took much longer to reach this threshold (8, 10, and 12 days, respectively). The HVEF-CP + nano-ZnO group only reached 6.25 and 6.13 lg (CFU·g^−1^) after 14 days at 4 °C and 10 days at 20 °C, respectively, demonstrating a significantly extended shelf life. This is consistent with the conclusions of Kongboonkird et al. [44].

#### 3.3.2. Effects of Different Sterilization Methods on TSS Content

As shown in Figure 12, the TSS content during storage at 4 °C and 20 °C first increased and subsequently decreased with time, reaching a maximum after 6 and 4 days, respectively, and gradually decreased after that. The TSS contents in each group after storage at 4 °C for 14 days and 20 °C for 10 days were 8.75% and 8.32% (control), 8.54% and 8.86% (PE film), 9.13% and 8.47% (nano-ZnO), 9.55% and 9.77% (HVEF-CP), and 10.33% and 10.12% (HVEF-CP + nano-ZnO), respectively. The initial increase in TSS content is due to fruit ripening, when grape hydrolase induces starch decomposition. Harvested grapes retain physiological activities such as respiration, and some substrates are consumed to provide energy. Simultaneously, the growth of microbes also consumes the substrates. With prolonged storage, the TSS content gradually decreased for each group. The propagation of microorganisms induces the depletion of nutrients and consequently a lower TSS content [45]. Nevertheless, the HVEF-CP + nano-ZnO group showed TSS contents of 10.33% on day 14 (4 °C) and 10.12% on day 10 (20 °C), which were significantly higher than those of the control group (*p* < 0.05) and higher than the other groups. Thus, the co-processing using HVEF-CP + nano-ZnO can effectively extend the storage time of grapes and slow down their nutrient loss, which is consistent with the findings by Sarangapani et al. [46].

#### 3.3.3. Effects of Different Sterilization Methods on VC Content

As shown in Figure 13, using either one sterilization method or both had no significant effect (*p* < 0.05) on the VC content of grapes compared to the control on day 0, regardless of the storage temperature. The VC content gradually decreased with time, especially at higher temperatures. After 14 days at 4 °C, the VC content (unit: mg·100 g^−1^) in the control group dropped from 5.65 to 2.82, that in the PE film group from 5.65 to 2.86, that in the nano-ZnO group from 5.62 to 3.21, that in the HVEF-CP group from 5.63 to 3.37, and that in the HVEF-CP + nano-ZnO group from 5.64 to 3.52. After 10 days at 20 °C, the corresponding changes were 5.65 to 2.61, 5.65 to 2.65, 5.62 to 3.18, 5.63 to 3.31, and 5.64 to 3.55, respectively. Co-processing the grapes using both HVEF-CP and nano-ZnO antimicrobial film significantly prolonged the storage period and effectively slowed down nutrient loss. HVEF-CP produces active substances that have significant antibacterial effects and therefore slow down VC depletion due to microbial activity. At the same time, these substances enhance the activity of dehydroascorbic reductase, facilitating the regeneration of VC in grapes [47].

#### 3.3.4. Effects of Different Sterilization Methods on TA Content

As shown in Figure 14, the TA content showed no significant change during storage at either 4 °C or 20 °C. However, the HVEF-CP + nano-ZnO treatment was more effective in slowing down TA decline and maintaining the flavor quality compared to the other four groups [48]. The TA content in this group decreased from 0.49% to 0.44% after 14 days and to 0.46% after 10 days at both 4 °C and 20 °C, remaining significantly higher than the other groups. This result indicates that co-processing with HVEF-CP + nano-ZnO antimicrobial film not only extends the storage period but also reduces organic acid consumption by grape metabolism, thus preserving the flavor quality to the greatest extent. This is in agreement with the findings of Jia et al. [49].

#### 3.3.5. Effects of Different Sterilization Methods on Mass Loss

As shown in Figure 15, the mass loss rate gradually increased during storage at both 4 °C and 20 °C. During transportation and storage, continuous physiological activities such as respiration cause the loss of water and nutrients in grapes [50,51], especially at higher temperatures. Compared to the control, HVEF-CP, and nano-ZnO groups, the HVEF-CP + nano-ZnO group showed slower mass loss after 14 days at 4 °C and 10 days at 20 °C. However, the mass loss was still higher than that of the PE film group. The poor water permeability of the PE film creates a more humid environment, which slows water loss but unfortunately also facilitates microbial growth and thus reduces the storage time. This result is consistent with the findings discussed in Section 3.3.1. Overall, co-processing grapes with HVEF-CP and nano-ZnO antimicrobial film can prolong the storage period and reduce the loss of water and nutrients.

## 4. Conclusions

This study examined the antimicrobial performance of a combined approach (HVEF-CP + nano-ZnO) for fresh grapes for the first time. The effects of different treatments on grape quality and shelf life were also investigated. The lg (CFU·g^−1^) value of *S. aureus* on grapes sterilized with HVEF-CP is affected by the voltage, frequency, and time of processing. Using the response surface test, the optimal processing parameters were determined to be a voltage of 78 kV, a frequency of 110 Hz, and a time of 116 s, which gave the minimum lg (CFU·g^−1^) value of 2.43 and the maximum sterilization rate of 96.29%. The influences of these three parameters are ranked as voltage > time > frequency (*p* < 0.05). Treating grapes with HVEF-CP alone and nano-ZnO antimicrobial film alone resulted in a sterilization rate of 96.28% and 78.62%, respectively. The combination of these two methods gave a much higher sterilization rate of 99.34%, indicating a significant synergistic effect. When the co-processed grapes were stored at 4 °C and 20 °C, their shelf life was extended by 8 and 6 days, respectively. Thus, the co-processing not only effectively suppressed microbial growth but also delayed deterioration and helped preserve the flavor and quality of grapes. Due to its high efficiency, safety, and scalability, this novel co-processing approach facilitates further development and optimization of grape preservation technologies to meet the rising market demand. The results also offer useful technical reference for the non-thermal preservation of other perishable foods.

## Figures and Tables

**Figure 1 foods-12-04234-f001:**
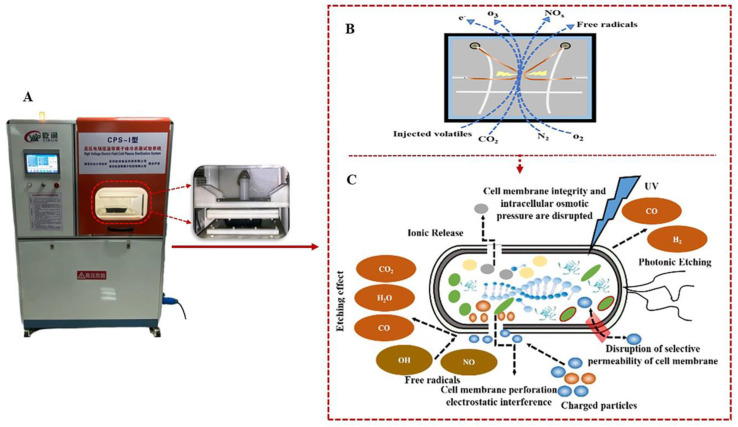
(**A**) Device for HVEF−CP generation. (**B**) Plasma formation mechanism. (**C**) Sterilization mechanism.

**Figure 2 foods-12-04234-f002:**
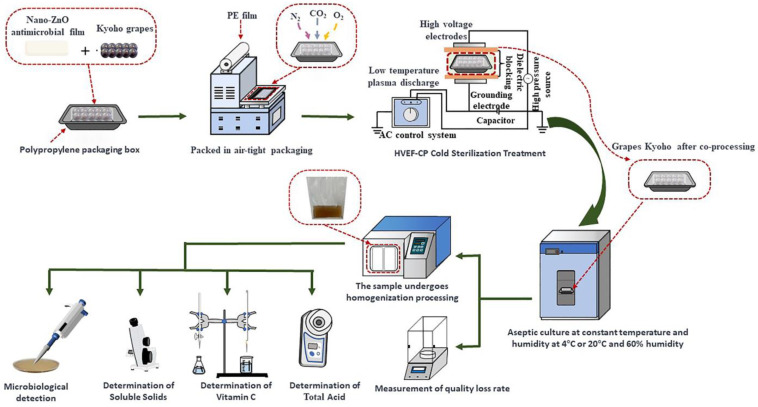
Process flowchart for synergistic treatment of grapes with HVEF-CP and nano-ZnO antimicrobial film.

**Figure 3 foods-12-04234-f003:**
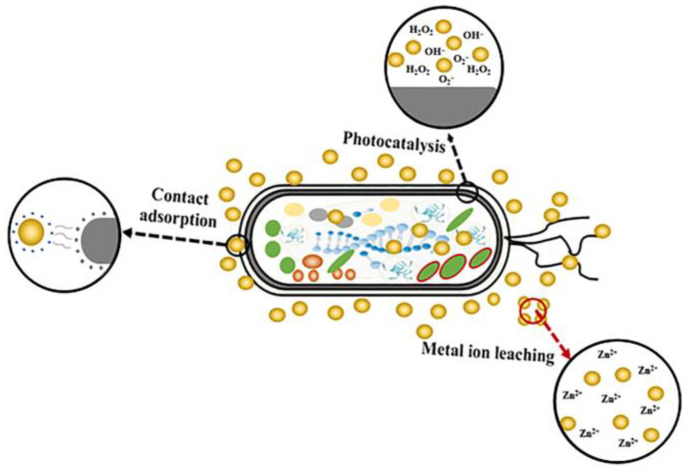
Antimicrobial mechanism of nano-ZnO.

**Figure 4 foods-12-04234-f004:**
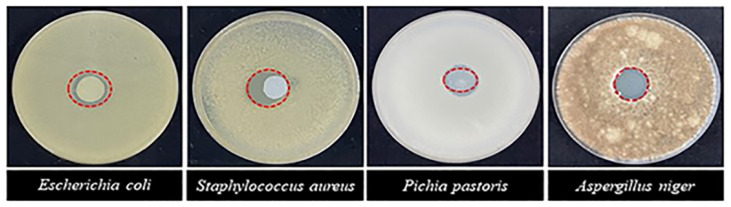
Bacteriostatic effect of nano-ZnO antimicrobial film. Diameter of the film disc: 10 mm.

**Figure 5 foods-12-04234-f005:**
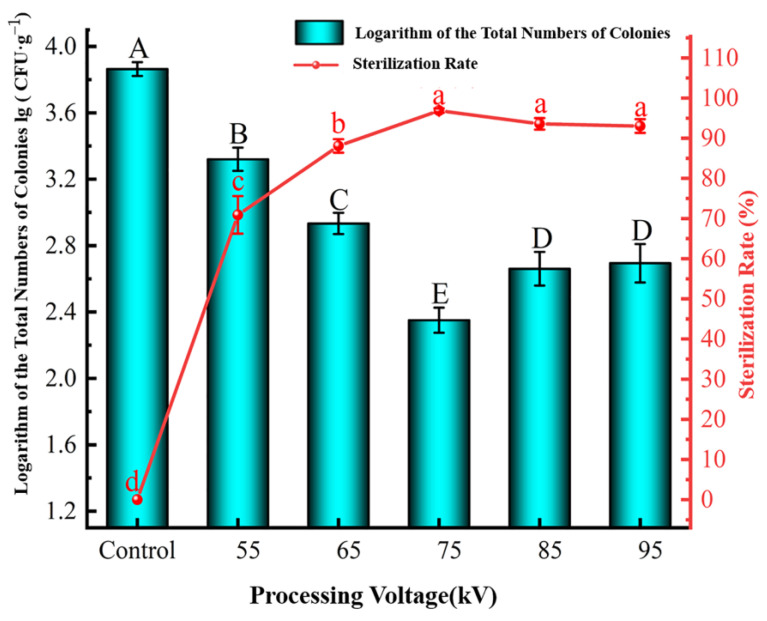
Effect of HVEF-CP processing voltage on the lg (CFU·g^−1^) value (left *y*-axis) and sterilization rate (right *y*-axis) of *S. aureus* at the grape surface.

**Figure 6 foods-12-04234-f006:**
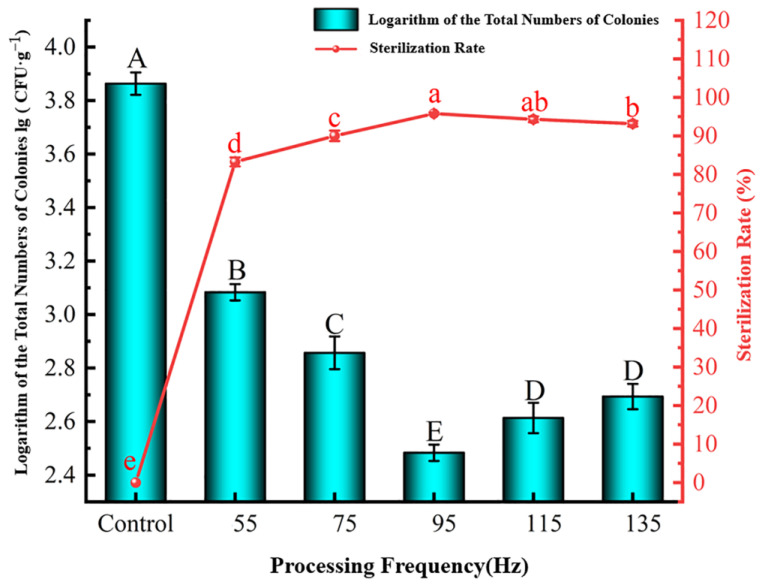
Effect of HVEF-CP processing frequency on the lg (CFU·g^−1^) value (left *y*-axis) and sterilization rate (right *y*-axis) of *S. aureus* at the grape surface.

**Figure 7 foods-12-04234-f007:**
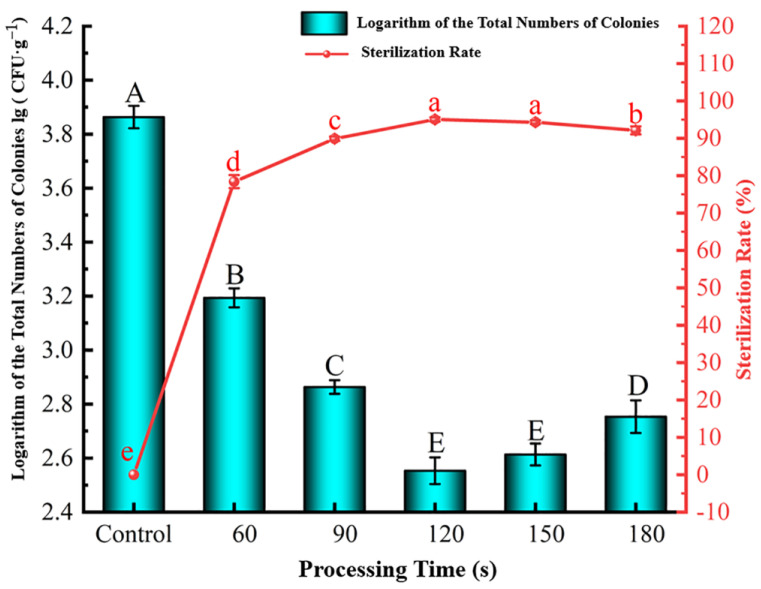
Effect of HVEF-CP processing time on the lg (CFU·g^−1^) value (left *y*-axis) and sterilization rate (right *y*-axis) of *S. aureus* at the grape surface.

**Figure 8 foods-12-04234-f008:**
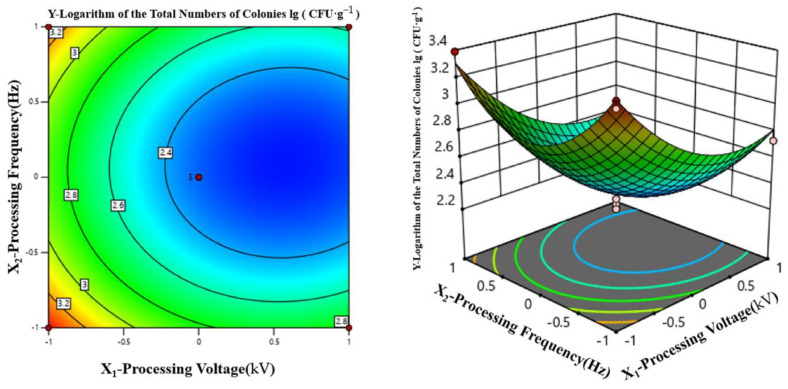
Joint effects of processing voltage and processing frequency on lg (CFU·g^−1^).

**Figure 9 foods-12-04234-f009:**
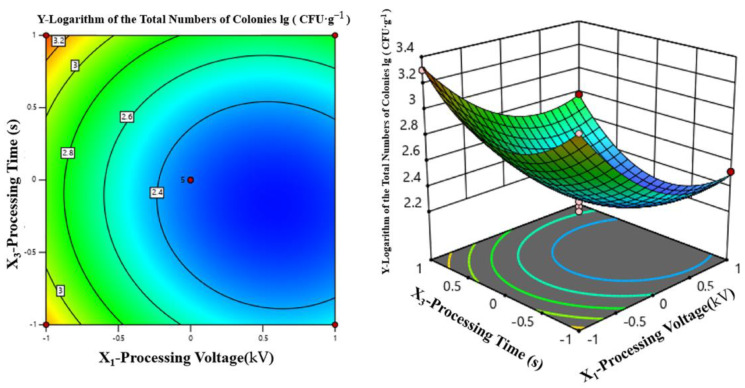
Joint effects of processing voltage and processing time on lg (CFU·g^−1^).

**Figure 10 foods-12-04234-f010:**
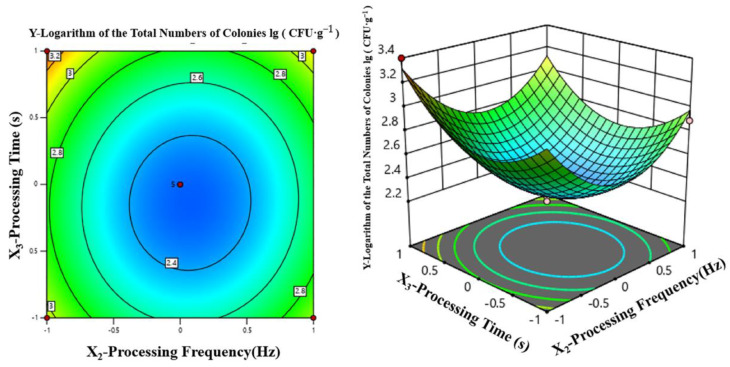
Joint effects of processing frequency and processing time on lg (CFU·g^−1^).

**Figure 11 foods-12-04234-f011:**
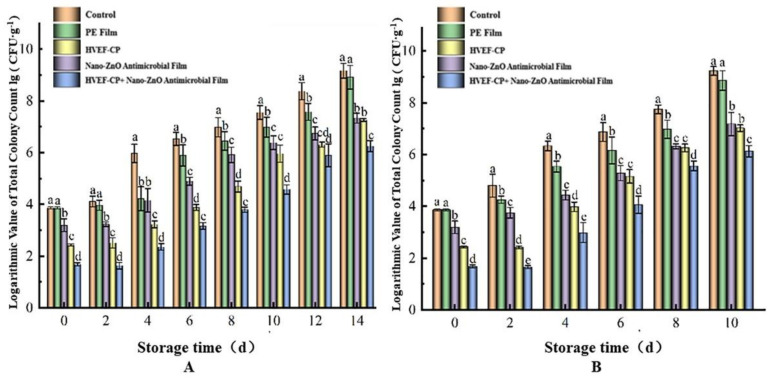
Effects of different treatments on the lg (CFU·g^−1^) value of grapes stored at (**A**) 4 °C and (**B**) 20 °C.

**Figure 12 foods-12-04234-f012:**
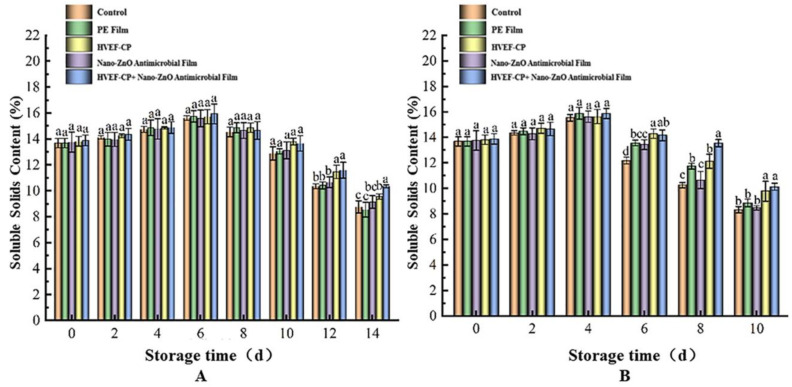
Effects of different treatments on the TSS content of grapes stored at (**A**) 4 °C and (**B**) 20 °C.

**Figure 13 foods-12-04234-f013:**
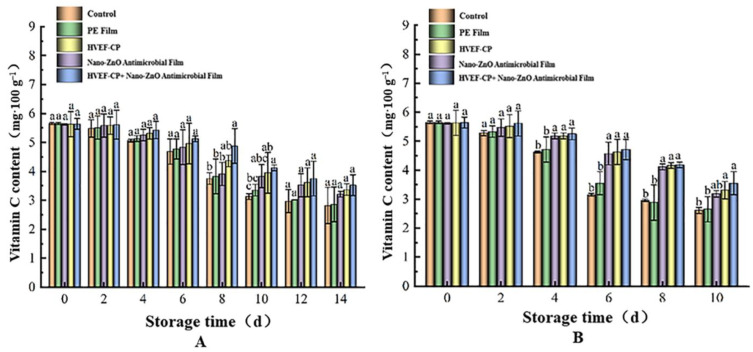
Effect of different treatments on the VC content of grapes stored at (**A**) 4 °C and (**B**) 20 °C.

**Figure 14 foods-12-04234-f014:**
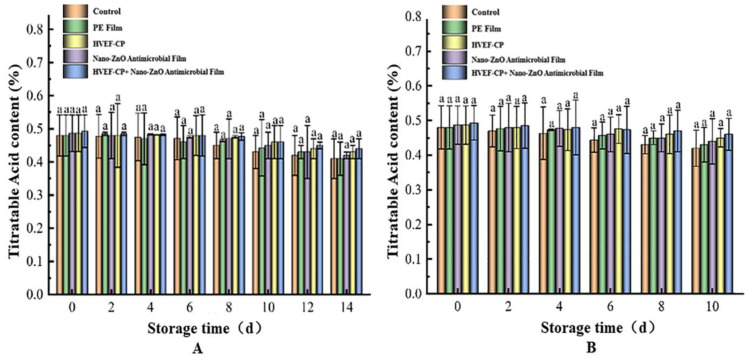
Effects of different treatments on the TA content of grapes stored at (**A**) 4 °C and (**B**) 20 °C.

**Figure 15 foods-12-04234-f015:**
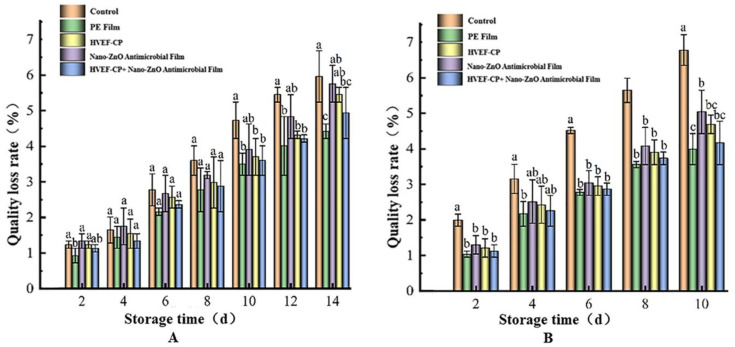
Effects of different treatments on the mass loss rate of grapes stored at (**A**) 4 °C and (**B**) 20 °C.

**Table 1 foods-12-04234-t001:** Factors and levels used in the single-factor experiments.

Level	Factor		
	Processing Voltage (kV)	Processing Frequency (Hz)	Processing Time (s)
1	55	55	60
2	65	75	90
3	75	95	120
4	85	115	150
5	95	135	180

**Table 2 foods-12-04234-t002:** Factors and levels used in the response surface design.

Factor	Level		
	−1	0	1
X_1_/processing voltage (kV)	65	75	85
X_2_/processing frequency (Hz)	75	95	115
X_3_/processing time (s)	90	120	150

**Table 3 foods-12-04234-t003:** Diameters of bacteriostatic ring after applying nano-ZnO antimicrobial film to different microbial species in a petri dish.

Microbial Species	Inhibition Zone Diameter (mm)
*Escherichia coli*	16.05 ± 0.56 ^b^
*Staphylococcus aureus*	18.57 ± 0.86 ^a^
*Pichia pastoris*	13.84 ± 0.72 ^c^
*Aspergillus niger*	13.96 ± 0.94 ^c^

Note: The diameter of the nano-ZnO antimicrobial film disc (10 mm) is included. All tests were performed in triplicate. Different superscript letters in the same column indicate significant differences, *p* < 0.05.

**Table 4 foods-12-04234-t004:** Box–Behnken design with experimental results.

Test Number	X_1_	X_2_	X_3_	lg (CFU·g^−1^)	
				Actual Value	Predicted Value
1	−1	−1	0	3.33	3.39
2	1	−1	0	2.73	2.82
3	−1	1	0	3.39	3.30
4	1	1	0	2.68	2.62
5	−1	0	−1	3.20	3.20
6	1	0	−1	2.52	2.50
7	−1	0	1	3.30	3.32
8	1	0	1	2.78	2.78
9	0	−1	−1	3.12	3.06
10	0	1	−1	2.89	2.97
11	0	−1	1	3.40	3.32
12	0	1	1	3.06	3.12
13	0	0	0	2.37	2.32
14	0	0	0	2.24	2.32
15	0	0	0	2.20	2.32
16	0	0	0	2.50	2.32
17	0	0	0	2.28	2.32

Note: All tests were performed in triplicate.

**Table 5 foods-12-04234-t005:** Analysis of variance for the fitted regression model.

Source	Sum of Squares	df	Mean Square	F Value	*p*-Value
Model	2.78	9	0.3093	21.25	0.0003 **
X_1_/processing voltage (kV)	0.7875	1	0.7875	54.12	0.0002 **
X_2_/processing frequency (Hz)	0.0392	1	0.0392	2.69	0.1447
X_3_/processing time (s)	0.0820	1	0.0820	5.64	0.0493 *
X_1_X_2_	0.0030	1	0.0030	0.2079	0.6622
X_1_X_3_	0.0064	1	0.0064	0.4398	0.5284
X_2_X_3_	0.0030	1	0.0030	0.2079	0.6622
X_1_^2^	0.3150	1	0.3150	21.65	0.0023 **
X_2_^2^	0.8189	1	0.8189	56.28	0.0001 **
X_3_^2^	0.5411	1	0.5411	37.19	0.0005 **
Residual	0.1019	7	0.0146		
Lack of fit	0.0446	3	0.0149	1.04	0.4663
Pure error	0.0573	4	0.0143		
Cor total	2.89	16			
CV = 4.27%					
R^2^ = 0.9647					
R^2^Adj = 0.9193					

Note: ** highly significant, *p* < 0.01; * significant, *p* < 0.05.

## Data Availability

Data are not available due to further project advancement.

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
