# Peer review of "Synergistic Microbial Inhibition and Quality Preservation for Grapes through High-Voltage Electric Field Cold Plasma and Nano-ZnO Antimicrobial Film Treatment"

_foods, 2023, doi:10.3390/foods12234234_

Round 1

Reviewer 1 Report

Comments and Suggestions for Authors

Review: „Innovative Technologies in Future Food Engineering: Synergistic Microbial Inhibition and Quality Preservation for Grapes Through High-Voltage Electric Field Cold Plasma and Nano-ZnO Antimicrobial Film Treatment”

The authors describe and statistically evaluate the outcome of experiments where grapes undergo a plasma treatment with a cold plasma. The authors investigate the antimicrobial impact of a plasma treatment in dependency of the voltage applied to the plasma source, the driving frequency of the plasma source, and the treatment time. The results have been statistically evaluated by a commercially available program. The manuscript is relatively well written and only need minor English checks. However, as I am not a native speaker, I do not feel competent enough to decide on language issues.

Remarks:

Material and methods: Your methods used in the experiments are not very well described. With your given information, the experiments are not reproducible. Please give information such as the temperatures for incubation, times, treatment times, or gasses used for your process. Please explain your statistics in detail. What is behind the program you are using. Design Expert 13, what is that program giving you? Is it freeware or commercial? Give a citation. Make your work reproducible.

Line 170/Figure 5: Please mention the microorganism used in that experiment in the caption of the figure. It is much easier to follow your thoughts and theories. Please follow that advice throughout all your figures. How many repetitions (N) do you have for your experiments? Your statistical statement is highly sensitive for that!

Line 248 – line 252: I do not understand this chapter. Please give more information.

Table 5 and general statements on your statistics: Please give more information about your statistics. Please highlight clearly what you did for your data evaluation. For instance, in Tab. 5 an F-test is mentioned. Since you are using SPSS for your evaluation, a Levene-test (a certain kind of variance test, https://www.youtube.com/watch?v=O6taUlWejB0) might be listed as an F-test, too. Which test did you use? Avoid misunderstandings and misinterpretations. However, are your variables (x1, x2, and x3) independent from each other? What are the influences on your model? Are there any changes when they are combined, meaningful for the model, etc.? Please mention! Discuss Tab. 5 in detail or give the reader an overview of the variable’s significance for the model. Did you do multiple tests, which would entail an adjustment of the significance level? Give all the basic information such as the kind of tests used, check for condition fulfillment (+ used tests for condition checks) in case of parametric tests, or the classification for the degree of significance (please use only one expression to describe (extremely/highly) in the chapter material and methods. However, well done!

I recommend to publish the manuscript in foods after major revisions.

Comments on the Quality of English Language

. The manuscript is relatively well written and only need minor English checks. However, as I am not a native speaker, I do not feel competent enough to decide on language issues.

Reviewer 2 Report

Comments and Suggestions for Authors

Dear Authors,

The manuscript entitled 'Innovative Technologies in Future Food Engineering: Synergistic Microbial Inhibition and Quality Preservation for Grapes Through High-Voltage Electric Field Cold Plasma and Nano-ZnO Antimicrobial Film Treatment' presents experimental results about the combination of two processes, one involving electric discharges, and one that involves ZnO nanoparticles, to effectively preserve the harvested grape crops. The authors claim that they have optimized the plasma treatment process (optimal parameters) to achieve a high degree of sterilization of grapes, 96%, and using additional zinc oxide nanoparticles, an efficiency of 99% has been reached.

The novel technique taken by the authors shows promise in that it has the potential to be effective, safe, and scalable when it comes to the preservation of grapes and other food items.

 The manuscript is carefully written, with concise and correctly positioned explanations in the text. The inserted images are explained and mentioned in the text, also having sufficient quality to be understood by the reader.

 The discussions are consistent and supported by the experimental results obtained by the authors. The conclusions are compact, perhaps too compact. A slight expansion of the conclusions section would be desirable.

Please include the corresponding microorganism types in figures 5 (on page 7), figure 6 (on page 7), and figure 7 (on page 8) as mentioned in the text. Additionally, intervention occurs as a result of the manipulation of microorganisms and the application of statistical methods. Furthermore, the choice of cohort for the experiment and the repeating of experimental trials also contribute to the intervention process. The inclusion of this information in the text is crucial, as it should be appropriately placed within the correct parts, rather than being omitted from the original edition of the document. In order to ensure transparency and reproducibility, it is imperative for authors to provide explicit details regarding the software solution employed for data analysis and statistical processing. This includes specifying the program utilized, its version, and any methods or options employed within the software to obtain the processed data that is presented in the study. The presence of this information must be established a priori, a detail that is absent in the original document.

I propose in a modified form with the few suggestions above.

Minor review.

Reviewer 3 Report

Comments and Suggestions for Authors

The research article by Li et al. investigated the synergistic effect of High-Voltage Electric Field Cold Plasma and Nano ZnO Antimicrobial Film treatment to ensure the quality and safety of harvested grapes. The research is well-developed, and the results are well-presented in this study. Some minor changes must be made before further processing. 

Minor:

Line 33. Please do not use “etc.” for scientific writing.

Lines 39-42. Please add a recent reference: https://doi.org/10.3390/app11020833

Lines 49-51. Please remove this sentence and Figure 1 from the Introduction.

Line 52. Please remove the word “also” and add a reference.

Lines 99-100. Please rewrite this sentence in the third person to improve soundness.

Line 101. Please revise everywhere lg to log.

Line 107. Please mention the duration.

Figure 2. Figure 2 can go into the Supplementary material.

Lines 160-161. How did the authors conclude the high/moderate etc. sensitivity of bacteria and fungi?

Lines 162-163. Was the size of the disc removed from the inhibition zone's diameter? Please do it and report the final inhibition zone diameter.

Figure 4. I assume drops of the formulation were added in the middle of the plates. To reduce the number of Figures and because these experiments are not very well performed (uneven disks), I suggest removing Figure 4 or adding that in the Supplementary materials.

Line 179. Please substitute “rose” with “increased”. Also, please avoid using adverts like “slightly” in the results section. Only mention if it was statistically significant or not.

Line 225. Please consider adding a short discussion in this section (3.2.2.1.) or combining sections 3.2.2.1 to 3.2.2.3 in one.

Lines 292. Please rewrite the Title.

Lines 295-296. Please rephrase. The authors could write instead that “Consuming food products contaminated with a high population of E. coli (6 log CFU/g) can cause human illness”.

Line 301. Please replace “very slow”.

Line 306. Please mention how much the shelf life was extended. Moreover, please mention the spoilage level of fresh fruits to determine the shelf life.

Conclusions. Please remove the unnecessary results from the conclusions and combine the main conclusions in one paragraph. 

Round 2

Reviewer 1 Report

Comments and Suggestions for Authors

The authors really improved their manuscript. It came out a fruitful discussion and from my point of view, nothing stands in the way for publishing the manuscript.

Comments on the Quality of English Language

I´m not a mother tongue. But the manuscript is well written and easy to read. Only minor English checks are necessary.